# Modeling and prediction of the 2019 coronavirus disease spreading in China incorporating human migration data

**Choujun Zhan[1], Chi K. Tse[2]\*, Yuxia Fu[3], Zhikang Lai[3], Haijun Zhang[4]**

**1** School of Computing, South China Normal University, Guangzhou, China, **2** Department of Electrical Engineering, City University of Hong Kong, Kowloon, Hong Kong, **3** School of Computer Science and Engineering, Nanfang College of Sun Yat-Sen University, Guangzhou, China, **4** Shenzhen Graduate School, Harbin Institute of Technology, Shenzhen, China

\* chitse@cityu.edu.hk

## Abstract

This study integrates the daily intercity migration data with the classic Susceptible-Exposed-Infected-Removed (SEIR) model to construct a new model suitable for describing the dynamics of epidemic spreading of Coronavirus Disease 2019 (COVID-19) in China. Daily intercity migration data for 367 cities in China were collected from Baidu Migration, a mobile-app based human migration tracking data system. Early outbreak data of infected, recovered and death cases from official source (from January 24 to February 16, 2020) were used for model fitting. The set of model parameters obtained from best data fitting using a constrained nonlinear optimisation procedure was used for estimation of the dynamics of epidemic spreading in the following months. The work was completed on February 19, 2020. Our results showed that the number of infections in most cities in China would peak between mid February to early March 2020, with about 0.8%, less than 0.1% and less than 0.01% of the population eventually infected in Wuhan, Hubei Province and the rest of China, respectively. Moreover, for most cities outside and within Hubei Province (except Wuhan), the total number of infected individuals is expected to be less than 300 and 4000, respectively.

## 1 Introduction

The Novel Coronavirus Disease 2019 (COVID-19) (known earlier as New Coronavirus Infected Pneumonia) began to spread since December 2019 from Wuhan, which has been widely regarded as the epicenter of the epidemic, to almost all provinces throughout China and 200 other countries. Up to July 23, 2020, a total of 15,416,529 cases of COVID-19 infection have been confirmed in 213 countries, and the death toll has reached 631,177. In the early phase of the outbreak, China was almost the only country affected by the virus, and on February 19, 2020 (when this work was completed), a total of 74,579 cases were confirmed in China, and the death toll was 2,119. Moreover, as human-to-human transmission had been found to

**Data Availability Statement:** Data of intercity traffic are all from Baidu Migration which is openly available: https://qianxi.baidu.com/. Data of COVID-19 infections are publicly available from National

Health Commission of China: http://www.nhc.gov.cn/xcs/yqtb/list_gzbd.shtml.

**Funding:** CZ was supported by the National Science Foundation of China Project 61703355 (http://www.nsfc.gov.cn/) and the Science and Technology Program of Guangdong 201904010224. CKT was supported by the City University of Hong Kong under Special Fund 9380114 (https://www.cityu.edu.hk/).

**Competing interests:** The authors have declared that no competing interests exist.

occur in some early Wuhan cases in mid December [1], the high volume and frequency of movement of people from Wuhan to other cities and between cities was an obvious cause for the wide and rapid spread of the disease throughout the country. Studies also suggested strong correlation between the spreading of infectious diseases with intercity travel [2]. The Susceptible-Exposed-Infected-Removed (SEIR) model has traditionally been used to study epidemic spreading with various forms of networks of transmission which define the contact topology [3], such as scalefree networks [4–6], small-world networks [7, 8], Oregon graph [9, 10], and adaptive networks [11]. Moreover, in most studies, the contact process assumes that the contagion expands at a certain rate from an infected individual to his/her neighbor, and that the spreading process takes place in a single population (network). The COVID-19 outbreak, however, began to occur in China and escalated in a special holiday period (about 20 days surrounding the Lunar New Year), during which a huge volume of intercity travel took place, resulting in outbreaks in multiple regions connected by an active transportation network. Thus, in order to understand the early transmission process of COVID-19 in China, it was essential to examine the human migration dynamics, especially between the epicenter Wuhan and other Chinese cities. Recent studies have also revealed the risk of transmission of the virus from Wuhan to other cities [12].

In this paper, we utilized the human migration data collected from Baidu Migration [13], which provided historical indicative daily volume of travellers to/from and between 367 cities in China [14, 15]. To demonstrate the impact of intercity traffic on the COVID-19 epidemic spreading, we plot in Fig 1 the number of infected individuals in different cities versus the inflow traffic volume from Wuhan, which clearly shows that for cities farther away from Wuhan, the number of infected individuals almost increases linearly with the inflow traffic from Wuhan. In view of the importance of human migration dynamics to the disease spreading process, we combine, in this study, intercity travel data collected from Baidu Migration [13] with the traditional SEIR model [3] to build a new dynamic model for the spreading of COVID-19 in China. Using official historical data of infected, recovered and death cases in 367 cities, we performed fitting of the data to estimate the best set of model parameters, which were then used to estimate the number of individuals exposed to the virus in each city and to predict the extent of spreading in the coming months. It should be noted that since January 24, 2020, very strict migration control had been imposed in various provinces and cities to restrict travel and hence to curb the spreading of the virus. Based on the early data, our study showed that provided such migration control and other stringent measures continued to be in place, the number of infected cases in various Chinese cities would peak between mid February to early March 2020, with about 0.8%, less than 0.1% and less than 0.01% of the population eventually infected in Wuhan, Hubei Province, and the rest of China, respectively, and no new cases to be expected from mid March. Moreover, for most cities in and outside Hubei Province (except Wuhan), the total number of infected individuals would be less than 4000 and 300, respectively. Finally, as the effectiveness of treatment improved, the recovery rate should increase and the epidemic in China was expected to end by June 2020. It should be stressed that our prediction, completed on February 19, 2020, used the early and relatively small amount of data, and thus verified effectiveness of the model using limited initial outbreak data in predicting pandemic progression.

In the remainder of the paper, we first introduce the official daily infection data and the intercity migration data used in this study. The SEIR model is modified to incorporate the human migration dynamics, giving a realistic model suitable for studying the COVID-19 epidemic spreading dynamics. Historical data of infected, recovered and death cases from official source and data of daily intercity traffic (number of travellers between cities) extracted from Baidu Migration were used to generate the model parameters, which then enabled estimation

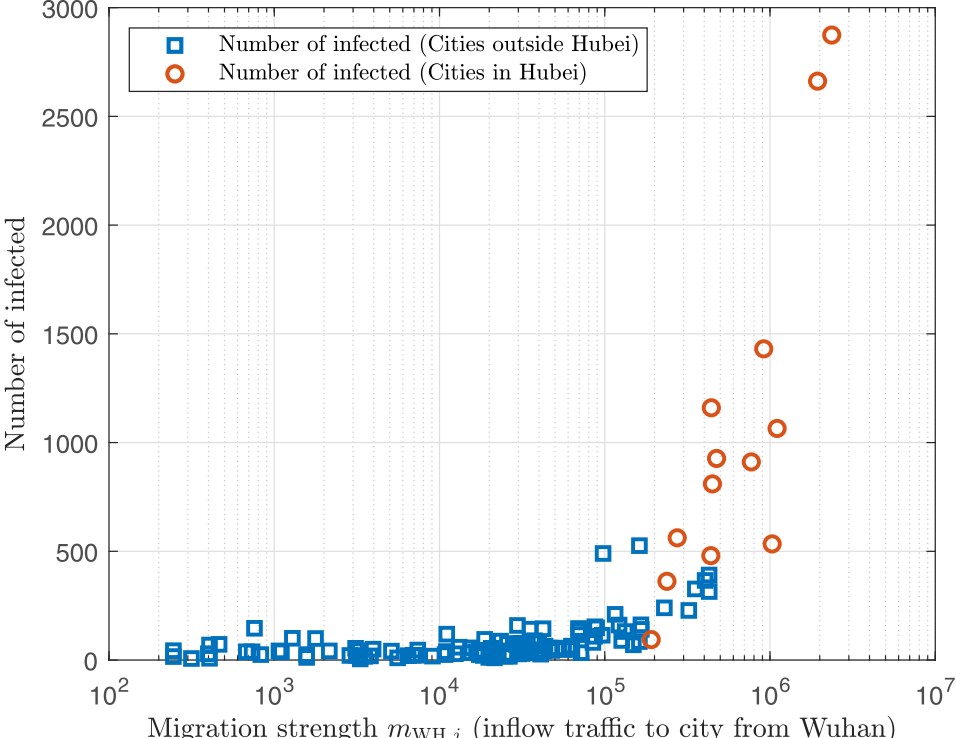

**Fig 1. Number of infected individuals in various cities on February 13, 2020 versus the city's inflow traffic from Wuhan.** Inflow traffic of each city from Wuhan is quantified by migration strength from Wuhan extracted from Baidu Migration data.

of the propagation of the epidemic in the following months. We will conclude with a brief discussion of our estimation of the propagation and the reasonableness of our estimation in view of the measures taken by the Chinese authorities in controlling the spreading of this new disease.

## 2 Data

### 2.1 Official data of COVID-19 cases

The availability of official data of infected cases in China varies from city to city. Wuhan, being the epicenter, had the first officially confirmed case of COVID-19 infection in China on December 8, 2019 [1]. Most other cities in China began to report cases of COVID-19 infections around mid January 2020. Our data of daily infected and recovered cases, and death tolls, were based on the official data released by the National Health Commission of China, and the daily data used in our study were from January 24, 2020, to February 16, 2020, including the daily total number of confirmed cases in each city, daily total cumulative number of confirmed cases in each city, daily cumulative number of recovered cases in each city, and daily cumulative death toll in each city. It should be emphasized that the official data may not be the actual (true) data. Although the earliest confirmed case in China appeared on December 8, 2019, subsequent missing cases were expected to be significant in Hubei Province in the early stage of the epidemic outbreak. Systematic updates of infection data in other cities began after January

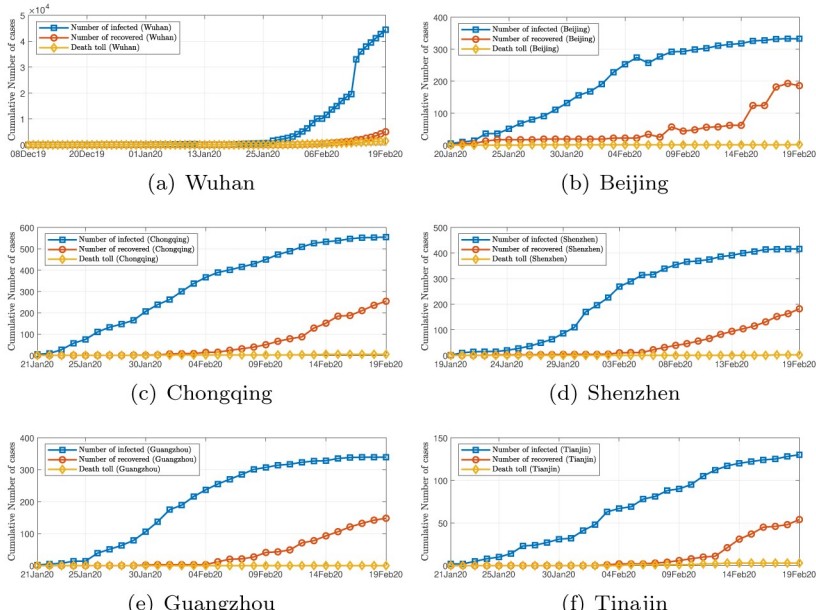

**Fig 2. Daily data of COVID-19 infections in six Chinese cities from December 8, 2019 to February 13, 2020.** (a) Wuhan (available from December 8, 2019); (b) Beijing (available from January 20, 2020); (c) Chongqing (available from January 20, 2020); (d) Shenzhen (available from January 19, 2020); (e) Guangzhou (available from January 21, 2020); (f) Tianjin (available from January 21, 2020).

17, 2020. Fig 2 shows the number of confirmed infected cases, recovered cases and death tolls of six major Chinese cities.

## 2.2 Intercity travel data

As human-to-human transmission had been confirmed to occur in the spreading of COVID-19, gatherings of people and intercity travel of infected and exposed individuals within China were identified as the main drives that escalated the spreading of the virus. The period (around 20 days) surrounding the Lunar New Year (mid January to early February in 2020) was the most important holiday period in China. Migrant workers and students traveled from major cities to country towns for family reunions, and returned to the cities at the end of the holiday period. Holiday goers also traveled to and from tourist cities. China's Ministry of Transport estimated around 3 billion trips to be taken during this period. Wuhan, being a major transport hub and having a large number of higher education institutions as well as manufacturing plants, was among the cities with the largest outflow and inflow traffic before and after the Chinese New Year festival. Our study aimed to incorporate these important human migration dynamics in the construction of the spreading model. We collected daily intercity travel data in China from Baidu Migration, which was a mobile-app based big data system recording movements of mobile phone users. Specifically, we collected Baidu Migration data for 367 cities (or administrative regions) in China over the period of January 1, 2020, to February 13, 2020. Moreover, Baidu Migration data were expected to be inexact and only indicative of the relative volume of movement of people from one city to another. Thus, the migration strengths of cities served as indicative measures of the human traffic volume moving in and out of individual cities and administrative regions, as depicted by the inflow and outflow networks

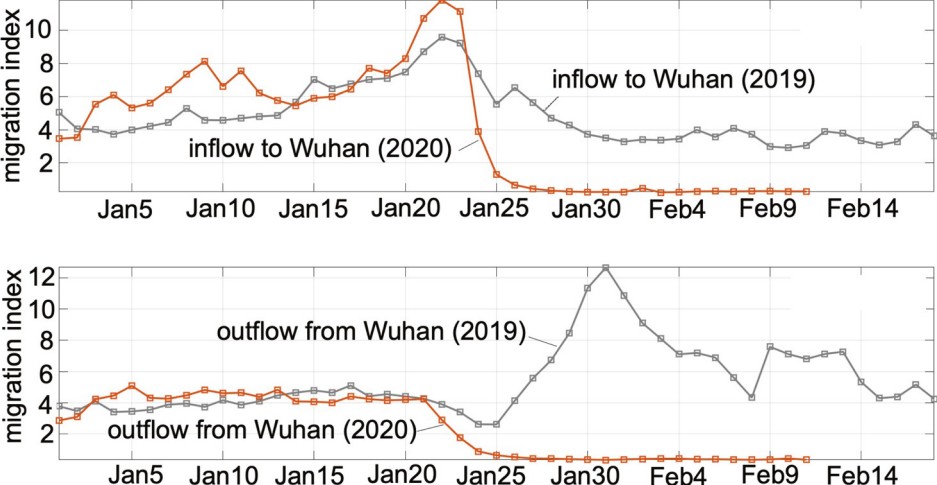

**Fig 3. Inflow and outflow data of each city with individual cities collected using Baidu Migration data.** Intercity migration strengths are used to form $m_{ij}$.

shown in Fig 3. Based on the collected data, we construct the *migration matrix*, i.e.,

$$M(t) = \begin{bmatrix} m_{11}(t) & m_{12}(t) & \cdots & m_{1K}(t) \\ m_{21}(t) & m_{22}(t) & \cdots & m_{2K}(t) \\ \vdots & \vdots & \ddots & \vdots \\ m_{N1}(t) & m_{N2}(t) & \cdots & m_{KK}(t) \end{bmatrix},$$ (1)

where $K$ is the number of the cities or administrative regions ($K = 367$ in this study), and $m_{ij}(t)$ is the migrant volume from city $i$ to city $j$ at time $t$. Migration matrix $M$ thus effectively describes the network of cities with human movement constituting the links of the network. as shown in Fig 3. Several properties of $M$ are worth noting:

- $M$ records migration from one city to another. Movement within a city is not counted, i.e., $m_{ii}(t) = 0$ for all $i$.

- $M$ is non-symmetric as traffic from one city to another is not necessarily reciprocal at any given time, i.e., $m_{ij}(t) \neq m_{ji}(t)$.

- Number of outflow migrants of city $i$ at time $t$ is

$$m_i^{(\text{out})}(t) = \sum_{i=j}^{K} m_{ij}(t).$$ (2)

- Number of inflow migrants of city $i$ at time $t$ is

$$m_i^{(\text{in})}(t) = \sum_{j=1}^{K} m_{ji}(t).$$ (3)

The right panels in Fig 3 plot the daily total inflow and outflow migration strengths of Wuhan, showing the abrupt decrease of migration strengths after the city shut down all inbound and outbound traffic from January 24, 2020.

## 3 Method

In the SEIR model, each individual in a population may assume one of four possible states at any time in the dynamic process of epidemic spreading, namely, susceptible (S), exposed (E), infected (I) and recovered/removed (R). The dynamics of the epidemic can be described by the following set of equations:

$$\dot{S}(t) = -\beta S(t)I(t), \dot{E}(t) = \beta S(t)I(t) - kE(t),$$

$$\dot{I}(t) = \kappa E(t) - \gamma I(t), \text{ and } \dot{R}(t) = \gamma I(t),$$

where $S(t)$, $E(t)$, $I(t)$ and $R(t)$ are, respectively, the number of people **s**usceptible to the disease, **e**xposed (being able to infect others but having no symptoms), **i**nfected (diagnosed as confirmed cases), and **r**ecovered (including death cases); $\beta$ is the exposition rate (infection rate of susceptible individuals); $\kappa$ is the infection rate of exposed individuals; and $\gamma$ is the recovery rate. For simplicity, recovered individuals include patients recovered from the disease and death tolls. In discrete form, the SEIR model can be represented by

$$\begin{aligned}
\Delta S(t) &= -\beta S(t-1)I(t-1), \\
\Delta E(t) &= \beta S(t-1)I(t-1) - \kappa E(t-1), \\
\Delta I(t) &= \kappa E(t-1) - \gamma I(t-1), \\
\Delta R(t) &= \gamma I(t-1)
\end{aligned} \tag{4}$$

where $\Delta S(t) = S(t) - S(t-1)$, $\Delta E(t) = E(t) - E(t-1)$, $\Delta I(t) = I(t) - I(t-1)$, and $\Delta R(t) = R(t) - R(t-1)$, with $t$ being a daily count. As the incubation period for COVID-19 can be up to 14 days, the number of exposed individuals (who show no symptom but are able to infect others) plays a crucial role in the spreading of the disease. The state $E$, which is not available from the official data, is thus an important state in our model. Furthermore, combining death toll with the recovered number as state $R$ will simplify the computation without affecting the accuracy of our data fitting and subsequent estimation.

### 3.1 Model

Suppose, for city $i$, the four states are $S_i(t)$, $E_i(t)$, $I_i(t)$ and $R_i(t)$, at time $t$. Here, we also define a total susceptible population, $N_i^s$, which is the eventual number of infected individuals in city $i$. Thus, $N_i^s$ represents the size of the group of susceptible, infected, exposed and removed individuals. Moreover, if city $i$ has a population of $P_i$ and the eventual percentage of infection is $\delta_i$, then $N_i^s = \delta_i P_i$. Thus, we have

$$N_i^s(t) = S_i(t) + E_i(t) + I_i(t) + R_i(t). \tag{5}$$

The classic SEIR model would give $\Delta I_i$ as the difference between the number of exposed individuals who become infected and the number of removed individuals. However, the onset of the COVID-19 epidemic has occurred in a special period of time in China, during which a huge migration traffic is being carried among cities, leading to a highly rapid transmission of the disease throughout the country. In view of this special migration factor, the SEIR model

should incorporate the human migration dynamics in order to capture the essential features of the dynamics of the spreading. In particular, for city $i$, in addition to the abovementioned classic interpretation, the daily increase in the number of infected cases should also include the inflow of infected individuals from other cities, less the outflow of removed cases from city $i$. In reality, inflow and outflow of exposed individuals to and from the city are also important and to be estimated in the model. Thus, if $m_{ij}(t)$ people move from city $i$ to city $j$ on day $t$, and the population of city $i$ is $P_i(t)$, then the number of infected individuals moving from city $i$ to city $j$ is

$$\Delta I_{ij}^{\text{in}}(t) = \frac{I_i(t)m_{ij}(t)}{P_i(t)}. \tag{6}$$

Also, the number of migrants leaving from city $j$ is $\sum_{i=1}^{N} m_{ji}$, and the number of infected cases that have migrated out of city $j$ is

$$\Delta I_j^{\text{out}}(t) = \frac{I_j(t) \sum_{i=1}^{N} m_{ji}(t)}{P_j(t)}, \tag{7}$$

where $P_j(t)$ is the population of city $j$ on day $t$. Thus, the increase in infected cases on day $t$ in city $j$ is given by

$$\begin{aligned}
\Delta I_j(t) &= \kappa_j(t)E_i(t) - \gamma(t)I_j(t) + \sum_{i=1}^{N} \Delta I_{ij}^{\text{in}}(t) - \Delta I_j^{\text{out}}(t) \\
&= \kappa_j(t)E_i(t) - \gamma_j(t)I_j(t) + \sum_{i=1}^{N} \left( \frac{I_i(t)m_{ij}(t)}{P_i(t)} \right) \\
&\quad - \frac{I_j(t) \sum_{i=1}^{N} m_{ji}(t)}{P_j(t)}
\end{aligned} \tag{8}$$

where $\Delta I_j(t) = I_j(t+1) - I_j(t)$ and $\kappa_j(t)$ is the infection rate in city $j$ on day $t$, i.e., the rate at which exposed individuals become infected. Moreover, infected individuals, once confirmed, would unlikely be able to migrate to another city. We thus implement this condition by writing (8) as

$$\begin{aligned}
\Delta I_j(t) &= \kappa_j(t)E_i(t) - \gamma_j(t)I_j(t) \\
&\quad + k_I \left( \sum_{i=1}^{N} \left( \frac{I_i(t)m_{ij}(t)}{P_i(t)} \right) - \frac{I_j(t) \sum_{i=1}^{N} m_{ji}(t)}{P_j(t)} \right)
\end{aligned} \tag{9}$$

where $0 < k_I \ll 1$ is a constant representing the possibility of an infected individual moving from one city to another.

Likewise, incorporating the migrant dynamics, the increase in exposed individuals on day $t$ in city $j$ is

$$
\begin{aligned}
\Delta E_j(t) \quad &= \frac{\beta_j(t)}{N_j^s(t)} I_j(t)S_j(t) + \frac{\alpha_j(t)}{N_j^s(t)} E_j(t)S_j(t) \\
&\quad - \kappa_j(t)E_i(t) + \sum_{i=1}^{N} \left( \frac{E_i(t)m_{ij}(t)}{P_i(t)} \right) \\
&\quad - \frac{E_j(t)\sum_{i=1}^{N} m_{ji}(t)}{P_j(t)}
\end{aligned}
\tag{10}
$$

where $\Delta E_j(t) = E_j(t+1) - E_j(t)$, $\beta_j$ is the infection rate of susceptible individuals in city $j$, and $\alpha_j$ is the infection rate of exposed individuals in city $j$. In a likewise fashion, we have

$$
\begin{aligned}
\Delta S_j(t) \quad &= -\frac{\beta_j(t)}{N_j^s(t)} I_j(t)S_j(t) - \frac{\alpha_j(t)}{N_j^s(t)} E_j(t)S_j(t) \\
&\quad + \sum_{i=1}^{N} \left( \frac{S_i(t)m_{ij}(t)}{P_i(t)} \right) - \frac{S_j(t)\sum_{i=1}^{N} m_{ji}(t)}{P_j(t)}
\end{aligned}
\tag{11}
$$

where $\Delta S_j(t) = S_j(t+1) - S_j(t)$. Finally, we have

$$
\Delta R_j(t) = \gamma_j(t)I_j(t),
\tag{12}
$$

where $\Delta R_j(t) = R_j(t+1) - R_j(t)$. In the above derivation, we should note that

- the recovered individuals are assumed to stay in city $j$;

- the recovery rates in different cities are assumed to be different due to varied quality of treatments and availability of medical facilities;

- the recovery rates increase as time goes, as treatment methods are expected to improve gradually (i.e., taking $\gamma_j(t)$ as a monotonically increasing function);

- the eventual recovery rates in all cities will converge to the same constant $\Gamma \approx 1$.

In addition, due to intercity migration, the population of city $j$ on day $t$ would increase or decrease according to

$$
\begin{aligned}
\Delta P_j(t) \quad &= \sum_{i=1}^{N} \left( \frac{P_i(t)m_{ij}(t)}{P_i(t)} \right) - \frac{P_j(t)\sum_{i=1}^{N} m_{ji}(t)}{P_j(t)} \\
&= \sum_{i=1}^{N} m_{ij}(t) - \sum_{i=1}^{N} m_{ji}(t)
\end{aligned}
\tag{13}
$$

where $\Delta P_j(t) = P_j(t+1) - P_j(t)$. Thus, the total susceptible population should be

$$\begin{aligned}
\Delta N_j^s(t) \quad &= k_I \left( \sum_{i=1}^{N} \left( \frac{I_i(t)m_{ij}(t)}{P_i(t)} \right) - \frac{I_j(t) * \sum_{i=1}^{N} m_{ji}(t)}{P_j(t)} \right) \\
&+ \sum_{i=1}^{N} \left( \frac{E_i(t)m_{ij}(t)}{P_i(t)} \right) - \frac{E_j(t) * \sum_{i=1}^{N} m_{ji}(t)}{P_j(t)} \\
&+ \sum_{i=1}^{N} \left( \frac{S_i(t)m_{ij}(t)}{P_i(t)} \right) - \frac{S_j(t) \sum_{i=1}^{N} m_{ji}(t)}{P_j(t)}
\end{aligned} \tag{14}$$

where $\Delta N_j^s(t) = N_j^s(t+1) - N_j^s(t)$.

In summary, our modified SEIR model with consideration of human migration dynamics, for city $j$, is given by

$$\begin{aligned}
\Delta I_j(t) \quad &= \kappa_j(t)E_i(t) - \gamma_j(t)I_j(t) \\
&+ k_I \left( \sum_{i=1}^{N} \left( \frac{I_i(t)m_{ij}(t)}{P_i(t)} \right) - \frac{I_j(t) * \sum_{i=1}^{N} m_{ji}(t)}{P_j(t)} \right), \\[6pt]
\Delta E_j(t) \quad &= \frac{\beta_j(t)}{N_j^s(t)} I_j(t)S_j(t) + \frac{\alpha_j(t)}{N_j^s(t)} E_j(t)S_j(t) \\
&- \kappa_j(t)E_i(t) + \sum_{i=1}^{N} \left( \frac{E_i(t)m_{ij}(t)}{P_i(t)} \right) \\
&- \frac{E_j(t) * \sum_{i=1}^{N} m_{ji}(t)}{P_j(t)}, \\[6pt]
\Delta S_j(t) \quad &= -\frac{\beta_j(t)}{N_j^s(t)} I_j(t)S_j(t) - \frac{\alpha_j(t)}{N_j^s(t)} E_j(t)S_j(t) \\
&+ \sum_{i=1}^{N} \left( \frac{S_i(t)m_{ij}(t)}{P_i(t)} \right) - \frac{S_j(t) \sum_{i=1}^{N} m_{ji}(t)}{P_j(t)}, \\[6pt]
\Delta R_j(t) \quad &= \gamma_j(t)I_j(t), \\[6pt]
\Delta P_j(t) \quad &= \sum_{i=1}^{N} m_{ij}(t) - \sum_{i=1}^{N} m_{ji}(t), \\[6pt]
\Delta N_j^s(t) \quad &= k_I \left( \sum_{i=1}^{N} \left( \frac{I_i(t)m_{ij}(t)}{P_i(t)} \right) - \frac{I_j(t) * \sum_{i=1}^{N} m_{ji}(t)}{P_j(t)} \right) \\
&+ \sum_{i=1}^{N} \left( \frac{E_i(t)m_{ij}(t)}{P_i(t)} \right) - \frac{E_j(t) * \sum_{i=1}^{N} m_{ji}(t)}{P_j(t)} \\
&+ \sum_{i=1}^{N} \left( \frac{S_i(t)m_{ij}(t)}{P_i(t)} \right) - \frac{S_j(t) \sum_{i=1}^{N} m_{ji}(t)}{P_j(t)}
\end{aligned} \tag{15}$$

where subscript $j$ denotes the city itself, and subscript $i$ denotes another city from/to which people migrate on day $t$. Letting $X_j(t)$ be the extended state vector, i.e.,

$X_j(t) = [S_j(t) \; E_j(t) \; I_j(t) \; R_j(t) \; P_j(t) \; N_j^s(t)]^T$, we write the above difference equation as

$$\Delta X_j(t) = f(X_j, X_i, \mu_i) \tag{16}$$

where $f(x)$ is the right side of (15), and $\mu_j$ is the set of parameters including $\alpha_j, \beta_j, \gamma_j, \kappa_j$ and $\delta_j$. For computational convenience, we write (15) as

$$X_j(t+1) = X_j(t) + f(X_j, X_i, \mu_i) \tag{17}$$

In performing the data fitting, we assume $\alpha_j(t), \beta_j(t), \gamma_j(t), \kappa_j(t)$, and $\delta_j$ are constants throughout the period of spreading, and the spreading begins at $t_0$, at which $N_j^s(t_0) = \delta_j P_j(t_0)$.

## 3.2 Parameter identification

The model represented by (17) describes the dynamics of the epidemic propagation with consideration of human migration dynamics. The parameters in model (17) are unknown and to be estimated from historical data. We solve this parameter identification problem via constrained nonlinear programming (CNLP), with the objective of finding an estimated growth trajectory that fits the data. An estimated number of infected cases of each city can be generated from (15) with unknown set $\theta_j$, i.e.,

$$\theta_j = \{\alpha_j, \beta_j, \gamma_j, \kappa_j, \delta_j, I_{j,0}\} \tag{18}$$

where $I_{j,0} = I_j(t_0)$ is the initial number of infections in city $j$, and $\{\alpha_j, \beta_j, \gamma_j, \kappa_j, \delta_j\}$ are parameters that determine the rates of spreading and recovery in city $j$. Then, the unknown set is $\Theta = \{\theta_1, \theta_2, \cdots, \theta_K\}$ essentially has $5K$ unknowns, where $K$ is the number of cities, thus requiring an enormous effort of computation. Here, to gain computational efficiency, we assume that

- all cities share one parameter set $\theta = \{\alpha, \beta, \kappa, \gamma\}$;

- the numbers of initial infected and exposed individuals in city $i$ are $\lambda_I I_i(t_0)$ and $\lambda_E I_i(t_0)$, respectively, where $\lambda_I$ and $\lambda_E$ are constant. Here, $I_i(t_0)$ represents the actual infected number at time $t_0$, while $\lambda_I I_i(t_0)$ represents the initial infection number used in the model;

- each city has an independent $\delta_i$.

  Then, the size of the unknown set becomes computationally manageable, i.e.,

$$\Theta = \{\alpha, \beta, \kappa, \gamma, \delta_i, \lambda_I, \lambda_E\}.$$

Finally, the parameter estimation problem can be formulated as the following constrained nonlinear optimisation problem:

$$P_0 : \min_{\Theta} \sum_{j=0}^{N} \|w_j(I(t_j) - \hat{I}(t_j))\|_l$$

$$s.t. \begin{cases} (i) & \hat{x}(t+1) = \hat{x}(t) + F(\hat{x}(t)), \\ \\ (ii) & \Theta_U \geq \Theta \geq \Theta_L, \end{cases} \tag{19}$$

where $F(\cdot)$ represents model (15) and $\hat{x}(t) = [\hat{I}(t), \hat{R}(t), E(t), S(t), P(t), N^s(t)]$ is the set of estimated variables, with unknown set $\Theta$, which is bounded between $\Theta_L$ and $\Theta_U$. In this work, an inverse approach is taken to find the unknown parameters and states by solving (19).

The Root Mean Square Percentage Error (RMSPE) is adopted as the criterion, i.e., fitting error, to measure the difference between the number of infected individuals generated by the

model and the official daily infection data.

$$\text{RMSPE} = \sqrt{\frac{1}{K}\sum_{i=1}\sum_{j=1}\left(\frac{\hat{I}_i(t_j) - I_i(t_j)}{I_i(t_j)}\right)^2} \times 100\%, \tag{20}$$

where $K$ is the number of cities to be evaluated.

## 4 Results

We perform data fitting of the model, described by (17), using historical daily infection data provided by the National Health Commission of China, from January 24, 2020 to February 13, 2020. Our approach, as described in the previous section, is to apply constrained nonlinear programming to find the best set of estimates for the unknown parameters and states. Data fitting for all 367 cities are performed. Values are updated iteratively in the optimisation process. Moreover, since all parameters, like infection rates, are to be generated by fitting data with the model, the integrity of the data becomes crucial. As the official Wuhan data are expected to deviate from the true values quite significantly during the early outbreak stage due to uncertainty in diagnosis and other issues related to reporting of the epidemic by the local government, we have allowed the fitting errors for Wuhan to expand over a reasonable range, while the fitting errors for most other cities remain small. In addition, as the epidemic propagates in time, effective control measures and improved public education would reduce the infection rates for the susceptible and exposed individuals, making these parameters time varying in reality. Nonetheless, our fitting assumes these parameters being constant during the short fitting period for computational simplicity.

The propagation profiles, in terms of the number of infected individuals and estimated number of exposed individuals, for all 367 cities are estimated. As limited by space, we only show in Fig 4 the results for 20 selected cities. This model can also provide projections of the number of infected and exposed individuals in the next 200 days, as shown in Fig 5, which clearly show that the daily infection would reach a peak sooner or later. By running the identification algorithm, we identified the optimal parameter set as $\alpha = 0.5869$, $\beta = 0.8949$, $\kappa = 0.1008$, $\gamma = 0.0602$, $\lambda_I = 1.9407$, and $\lambda_E = 1.5144$. From the estimated propagation profiles of the COVID-19 epidemic for all 367 cities, we have the following findings:

1. For most cities, the infection numbers would peak between mid February to early March 2020, as shown in Fig 6(a).

2. The peak number of infected individuals would be between 1,000 to 5,000 for cities in Hubei, and that outside Hubei would be below 500, as shown in Fig 6(b).

3. At the end, about 0.8%, less than 0.1% and less than 0.01% of the population would get infected in Wuhan, Hubei Province and the rest of China, respectively, as presented in Fig 6(c). Translating to actual figures, for most cities outside and within Hubei Province (except Wuhan), the total number of infected individuals was expected to be fewer than 300 and 4000, respectively, as shown in Fig 6(d).

4. For Wuhan, our model showed that the cumulative number of infections was 105,244 (95% CrI [64297, 146191]), which was consistent with a previous estimation of 75,815 cases (95% CrI [37304, 130330]) [16].

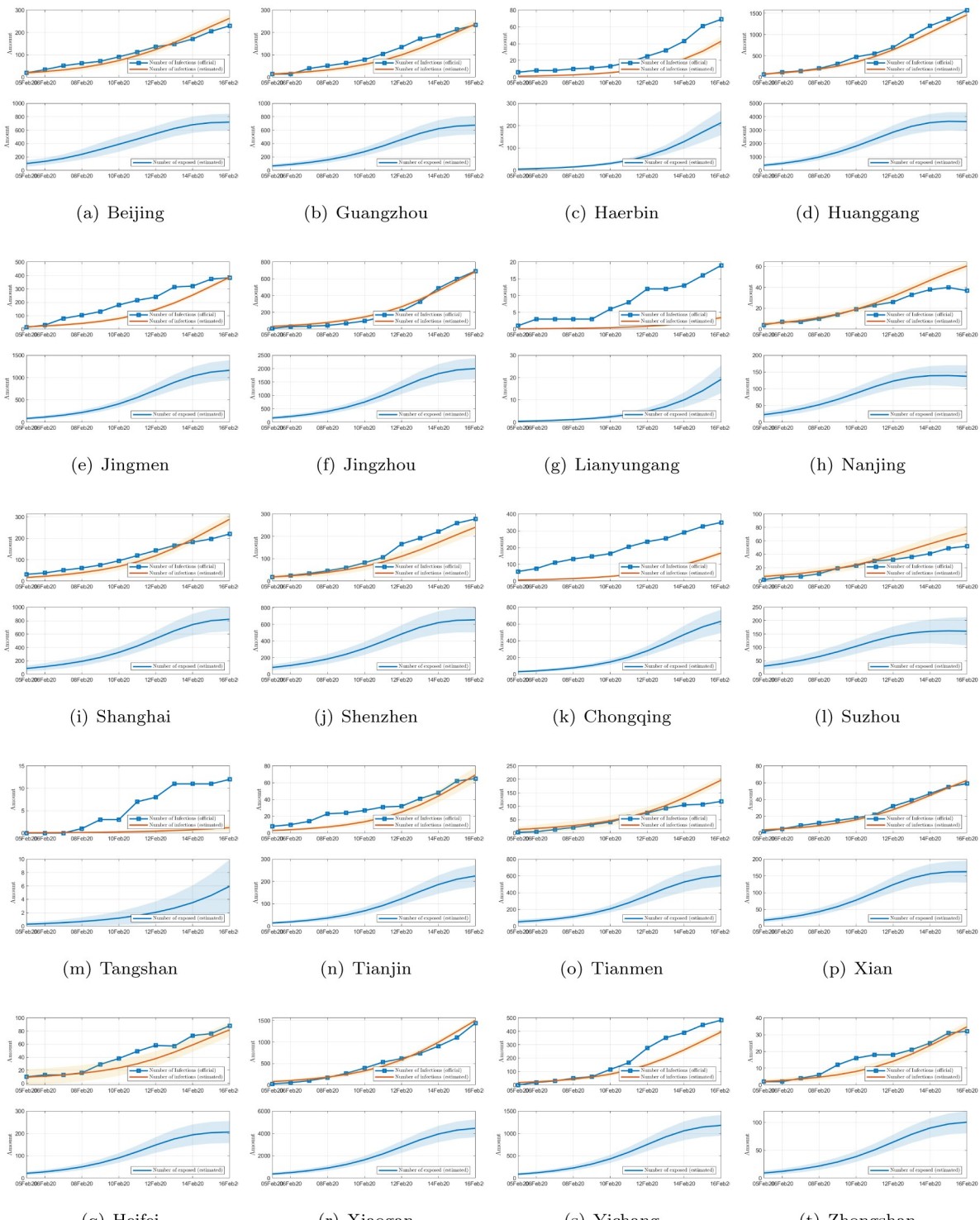

**Fig 4. Official number of infected individuals and estimated number of infected individuals in 20 selected cities in China (upper), and estimated number of exposed individuals (lower), while the filled area shows the 95% confidence interval.**

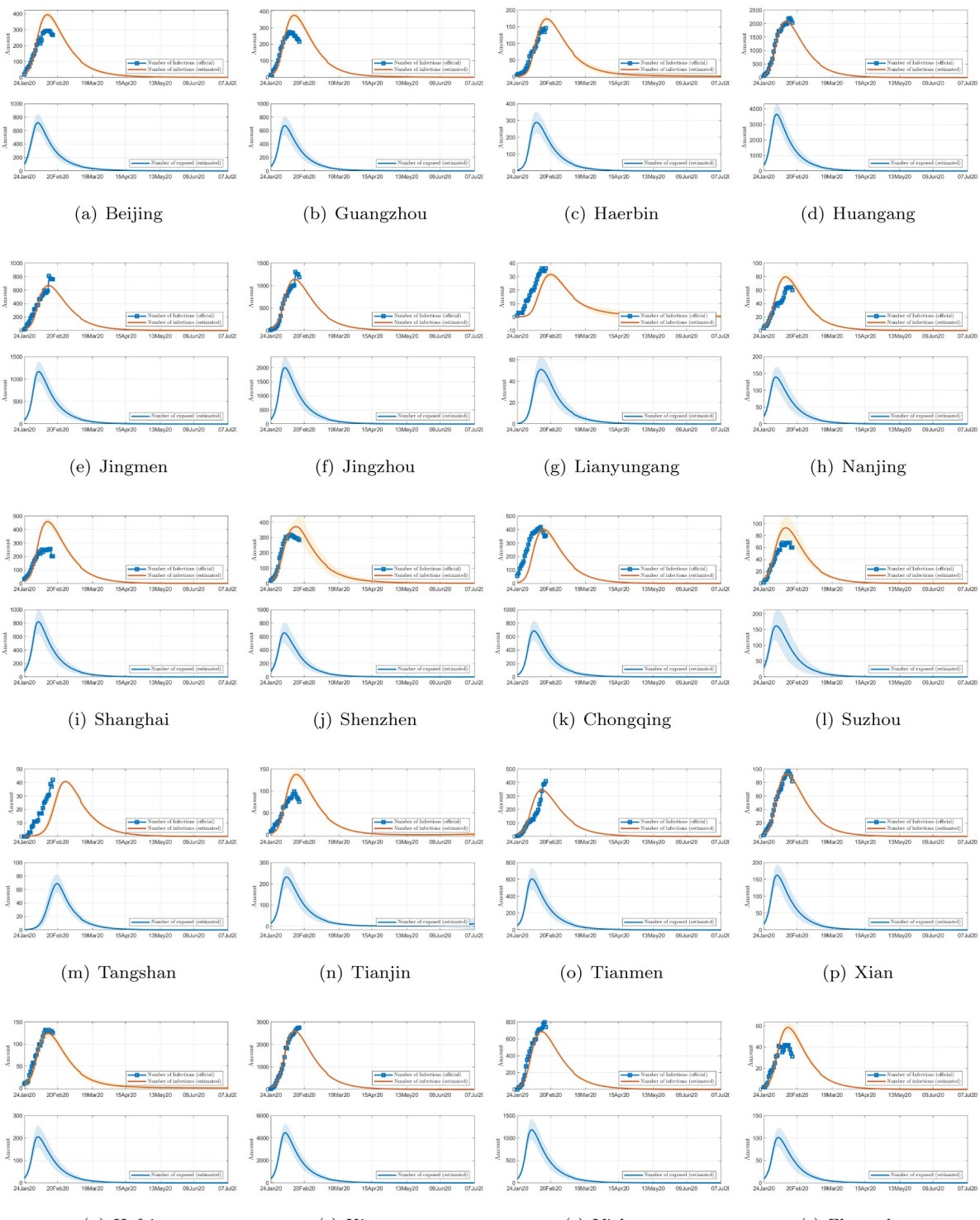

**Fig 5. Prediction of the number of infected (upper) and exposed individuals (lower) in 20 selected cites in China for the next 150 days.**
The shaded band is the 95% confidence interval.

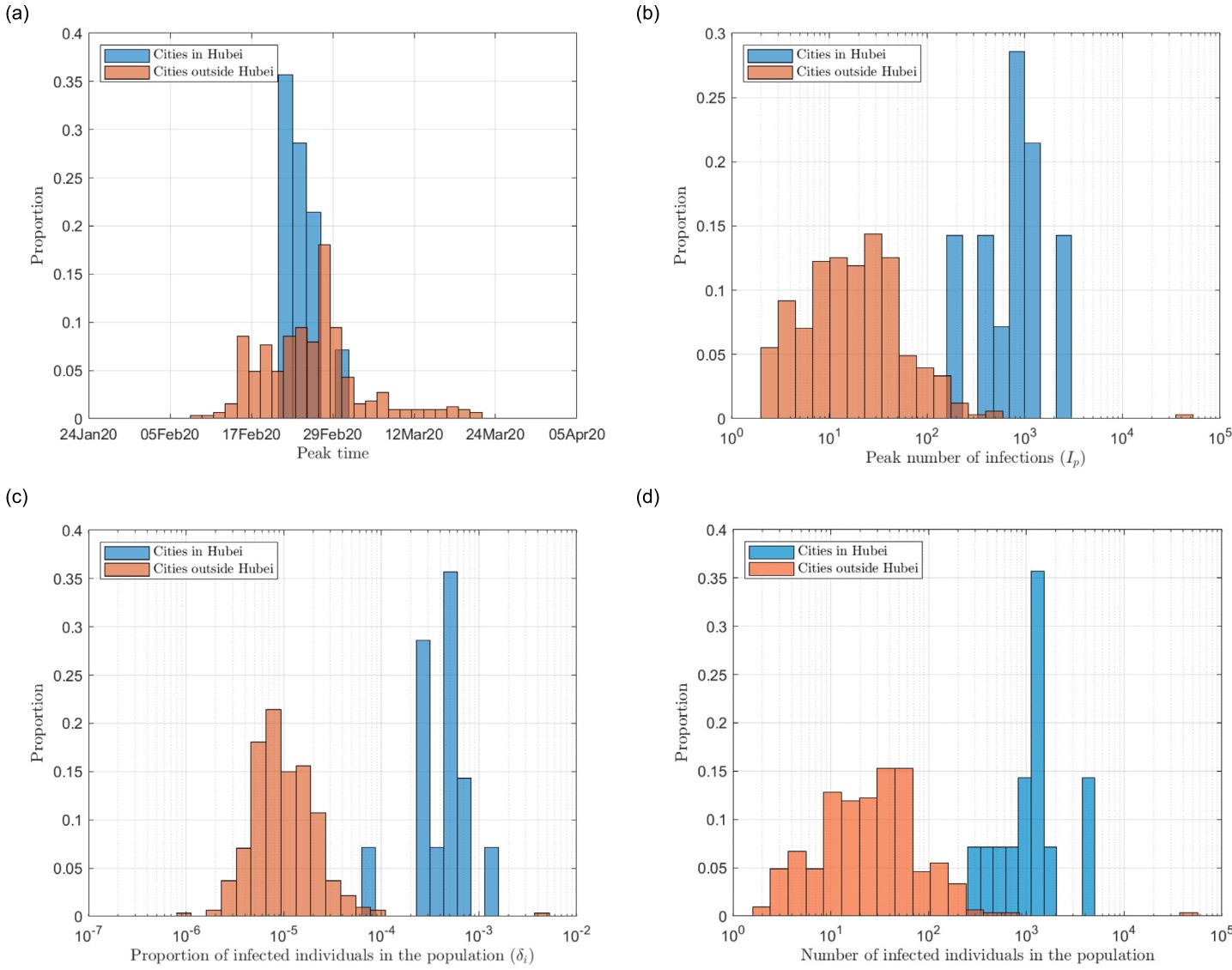

**Fig 6.** (a) Distribution of (a) peak time; (b) peak number of infections; (c) proportion of the population eventually infected in a city; (d) total number of the individuals eventually infected in a city.

## 5 Discussions

Opinions diverged on the estimated extent of the outbreak of the new coronavirus disease (COVID-19). While there were pure speculations, there were also predictions based on rigorous study of the spreading dynamics. Different models used for prediction and different assumptions made regarding the transmission process would lead to different results and quite diverged conclusions. For instance, an AI-powered simulation run had predicted 2.5 billion people to be infected in 45 days [17]. Academics in Hong Kong expected 1.4 million eventually infected in the city of 7.5 million people. Our results, however, did not seem to agree with such predictions. In fact, our results were expected to be optimistic, under normal circumstances, in the sense that the projected severity and duration of the epidemic were valid provided stringent measures continued to be in place to curb the spreading of the virus, especially before

mid March. Moreover, the effectiveness of medical treatment was expected to improve and the recovery rate was expected to increase in the following months. As our simulation was based on the data collected in the early outbreak phase, the recovery rate could be under-estimated. Should the recovery rate increase by 0.0005 each day, namely, the number of daily recovered individuals increases by 1% of the total number of infected individuals every 20 days, most cities in China would have zero infection case by June 2020. However, as the world is connected and unless strict travel bans were in place (currently most countries still allow their own citizens to return), possibility exists for infected individuals including those who are asymptomatic to move from city to city, however small in quantity. Second and third waves of outbreaks could not be ruled out! A high level of vigilance should be maintained to prevent the continuous spread of the virus, especially via the active transportation network. Furthermore, since this work was completed on February 19, 2020 (medRxiv 10.1101/2020.02.18.20024570), we used a short historical epidemic data and migration data to develop the model and the corresponding system identification algorithm. At the time of performing this work, there was no attempt in combining SEIR model, migration data and system identification techniques to analyze and predict the spread of COVID-19. The results thus have important indicative values on the effectiveness of using limited initial outbreak data in predicting pandemic progression.

## 6 Conclusion

The Novel Coronavirus Disease 2019 (COVID-19) epidemic has initially hit China hard. While the virus began to spread to other countries from February 2020, the extent of the outbreak in China remained to be severe in comparison to other countries for much of March and April 2020. Prediction of the severity and duration of the epidemic provided essential information for illuminating social and non-pharmaceutical interventions. However, prediction with the needed level of accuracy was a non-trivial task. In this work, we employed human migration data to provide information on intercity travel that was crucial to the transmission of the novel coronavirus disease from its epicenter Wuhan to other parts of China. The model described in this paper was essentially the classic SEIR model, with intercity travel data supplying the essential information about the number of infected, exposed and recovered individuals moving between different cities. All parameters of the model, including infection rates, recovery rates, and eventual percentage of infected population for 367 cities in China, were identified by fitting the official data collected up to mid-February with the model using a constrained nonlinear programming procedure. Using these parameters, predictions of the number of exposed individuals in 367 cities as well as projections into the next 200 days were made. Our model, however, did not consider the contact network topology that would be necessary if details of the transmission process, such as superspreading events, were to be captured. Nonetheless, our model provided a highly consistent estimation of the propagation of average numbers of exposed, infected and recovered individuals, despite missing details of fluctuation (e.g., sudden surge due to a superspreading event).

 Our prediction in mid February 2020 was that provided stringent control measures including travel restriction continue to be in place, the COVID-19 epidemic spreading would peak between mid February to early March 2020, with about 0.8%, less than 0.1% and less than 0.01% of the population eventually infected in Wuhan, Hubei Province and the rest of China, respectively. Moreover, as the effectiveness of treatment improved, the COVID-19 epidemic was expected to end by June 2020. However, possibilities of a second or third wave of outbreaks may exist as intercity travel is still permitted, e.g., homebound travel from regions which are still at different stages of the pandemic progression. It is thus advisable to maintain a

high level of vigilance by the public as well as a high level of preparedness for reactivating stringent control measures by government authorities.

## Author Contributions

**Conceptualization:** Chi K. Tse.

**Data curation:** Yuxia Fu, Zhikang Lai, Haijun Zhang.

**Formal analysis:** Choujun Zhan, Chi K. Tse, Zhikang Lai.

**Funding acquisition:** Choujun Zhan, Chi K. Tse.

**Investigation:** Chi K. Tse, Yuxia Fu, Zhikang Lai, Haijun Zhang.

**Methodology:** Choujun Zhan, Chi K. Tse.

**Software:** Choujun Zhan, Yuxia Fu, Zhikang Lai, Haijun Zhang.

**Supervision:** Chi K. Tse.

**Validation:** Yuxia Fu, Zhikang Lai, Haijun Zhang.

**Writing – original draft:** Choujun Zhan, Chi K. Tse.

**Writing – review & editing:** Chi K. Tse.

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
