## [Decision Letter · Decision Letter 0]

9 Jun 2020

PONE-D-20-05906

Modelling and Prediction of the 2019 Coronavirus Disease Spreading in China Incorporating Human Migration Data

PLOS ONE

Dear Dr. Tse,

Thank you for submitting your manuscript to PLOS ONE. After careful consideration, we feel that it has merit but does not fully meet PLOS ONE’s publication criteria as it currently stands. Therefore, we invite you to submit a revised version of the manuscript that addresses the points raised during the review process.

In particular, both reviewers would like the authors to include comparisons against similar studies already published, on the effects of human migration. Please also be sure to address Reviewer #2's comments on the infection starting in early November 2019 instead of during the Spring Festival in February 2020, and to use more current numbers than those available at the time of submission.

We look forward to receiving your revised manuscript.

Kind regards,

Siew Ann Cheong, Ph.D.

Academic Editor

PLOS ONE

Journal Requirements:

3. We note that Figure 3 in your submission contain map images which may be copyrighted. All PLOS content is published under the Creative Commons Attribution License (CC BY 4.0), which means that the manuscript, images, and Supporting Information files will be freely available online, and any third party is permitted to access, download, copy, distribute, and use these materials in any way, even commercially, with proper attribution. For these reasons, we cannot publish previously copyrighted maps or satellite images created using proprietary data, such as Google software (Google Maps, Street View, and Earth). For more information, see our copyright guidelines: http://journals.plos.org/plosone/s/licenses-and-copyright.

3.1.    You may seek permission from the original copyright holder of Figure 3 to publish the content specifically under the CC BY 4.0 license.

3.2.    If you are unable to obtain permission from the original copyright holder to publish these figures under the CC BY 4.0 license or if the copyright holder’s requirements are incompatible with the CC BY 4.0 license, please either i) remove the figure or ii) supply a replacement figure that complies with the CC BY 4.0 license. Please check copyright information on all replacement figures and update the figure caption with source information. If applicable, please specify in the figure caption text when a figure is similar but not identical to the original image and is therefore for illustrative purposes only.

4. Please update your submission to use the PLOS LaTeX template. The template and more information on our requirements for LaTeX submissions can be found at http://journals.plos.org/plosone/s/latex.

Reviewers' comments:

Reviewer's Responses to Questions

**Comments to the Author**

1. Is the manuscript technically sound, and do the data support the conclusions?

Reviewer #1: Partly

Reviewer #2: Yes

2. Has the statistical analysis been performed appropriately and rigorously? 

Reviewer #1: Yes

Reviewer #2: Yes

3. Have the authors made all data underlying the findings in their manuscript fully available?

Reviewer #1: No

Reviewer #2: Yes

4. Is the manuscript presented in an intelligible fashion and written in standard English?

Reviewer #1: Yes

Reviewer #2: Yes

5. Review Comments to the Author

Reviewer #1: This study used SEIR compartments to simulate the dynamics of COVID-19. It is an important issue now but there are some concerns as follows.

1. Many studies incorporated migration data into SEIR model for simulating epidemic dynamics. Authors need to highlight the significant findings of the study.

2. Recent related studies on modeling COVID-19 which simulating the dynamics Wuhan and Hubei for estimated infected persons were published. Authors need to add more comparisons with these studies.

3. Some notations need to be further clarify. For example, the notation of denominators in Equation 6 and Equation 7 should be P_i (t) and P_j (t), respectively.

4. If N_i^s (t) means the susceptible population in city i at time t, isn’t it similar to S_i (t)? please explain their difference more clearly.

5. Which is your notation of initial infection number? I_i (t_0) in line 228 or λ_I I_i (t_0) in line 236?

6. What is the fitting result of the following parameters: δ_i, λ_I, and λ_E?

7. Figure 5 displays the result of forecasting; it should add 95% confidence intervals or error bars to show the variations of estimated values.

8. What is the spatial variation of the prediction? For example, whether the cities strongly interacting with Wuhan have more precise prediction results than the other cities? Or, whether high-population-density cities have more accurate predictions? These comparisons may reflect the value of incorporating human migration data into a SEIR model so that model results can benefit real epidemic prevention tasks.

Reviewer #2: This is a sound analysis of two publicly available data set focusing on intercity migration in China.

The authors may benefit from aa recent paper on migration and covid-19 spread published in April issue of Migration Letters journal. That can be useful to better frame the context of this paper indicating wider link between human mobility and disease diffusion.

Authors may revisit the sentence in second page: "The COVID-19 outbreak, however, began

to occur and escalate in a special holiday period in China (about days surrounding the Lunar New Year), during which a huge volume of intercity travel took place, resulting in outbreaks in multiple regions connected by an active transportation network." Because now we know the virus was out and about in as early as early November.

The data needs to be critically presented; Authors indicate the possibility of incompleteness or inaccuracy of official covid data but it seems they assume Baidu data is free of problems. It is important to note the selectivity bias here. This data is collected by an app, which means there are a lot of questions about its representability. This should be clearly noted so readers can interpret it accordingly.

In the conclusion, authors state "The Coronavirus Disease 2019 (COVID-19) epidemic has hit China hard, 331 and as of February 20, 2020, a total of 74,579 infection cases have been 332 confirmed in China, with death toll reaching 2,119." It is a live incidence but it can be useful if they can include the latest statistics regarding the pandemic while making sure the data and analysis refer to an earlier period.

6. PLOS authors have the option to publish the peer review history of their article (what does this mean?). If published, this will include your full peer review and any attached files.

Reviewer #1: No

Reviewer #2: No

---

## [Author Response · Author response to Decision Letter 0]

24 Jul 2020

Reviewer: 1

Comments to the Author

Reviewer #1: This study used SEIR compartments to simulate the dynamics of COVID-19. It is an important issue now but there are some concerns as follows.

1. Many studies incorporated migration data into SEIR model for simulating epidemic dynamics. Authors need to highlight the significant findings of the study.

Authors’ Response: This work was completed on February 19, 2020 (medRxiv 10.1101/2020.02.18.20024570). We used a short historical epidemic spreading data and migration data to develop the model and the corresponding system identification algorithm. At the time of performing this work, there was no attempt in combining SEIR model, migration data and system identification techniques to analyze and predict the spread of COVID-19. The results thus have important indicative values on the effectiveness of using limited initial outbreak data in predicting pandemic progression. Remarks have been added to the Discussions section to highlight this. The main findings were listed in the Results section.

2. Recent related studies on modeling COVID-19 which simulating the dynamics Wuhan and Hubei for estimated infected persons were published. Authors need to add more comparisons with these studies.

Authors’ Response: The following information has been added to the Results section.

“For Wuhan, our model shows that the cumulative number of infections was 105,244 (95% CrI [64297,146191]), which was consistent with previous estimation of 75,815 infected cases (95% CrI [37304,130330]) [15]”.

[16] Wu JT, Leung K, Leung GM. Nowcasting and forecasting the potential domestic and international spread of the 2019-nCoV outbreak originating in Wuhan, China: a modelling study. The Lancet. 2020 Feb 29; 395(10225):689-97.

3. Some notations need to be further clarify. For example, the notation of denominators in Equation 6 and Equation 7 should be P_i (t) and P_j (t), respectively.

Authors’ Response: The notations have been revised.

4. If N_i^s (t) means the susceptible population in city i at time t, isn’t it similar to S_i (t)? please explain their difference more clearly.

Authors’ Response: Thank you for pointing this out. N_i^s represents the size of the group of susceptible, infected, exposed and removed individuals. Thus, we have N_is(t_0) = S(t_0)+E(t_0)+I(t_0)+R(t_0). This has been included in the Method section of revised paper.

5. Which is your notation of initial infection number? I_i (t_0) in line 228 or λ_I I_i (t_0) in line 236?

Authors’ Response: I_i (t_0) represents the actual infected number at time t_0, while λ_I I_i (t_0) represents the initial infection number used in the model. We have clarified this in the paper.

6. What is the fitting result of the following parameters: δ_i, λ_I, and λ_E?

Authors’ Response: The fitting result of δ_i, λ_I, and λ_E have been added, while Figure 6 (c) shows the distribution of \\delta_i.

7. Figure 5 displays the result of forecasting; it should add 95% confidence intervals or error bars to show the variations of estimated values.

Authors’ Response: The 95% confidence intervals (CrI) have been added to Figures 4 and 5, and in the text.

8. What is the spatial variation of the prediction? For example, whether the cities strongly interacting with Wuhan have more precise prediction results than the other cities? Or, whether high-population-density cities have more accurate predictions? These comparisons may reflect the value of incorporating human migration data into a SEIR model so that model results can benefit real epidemic prevention tasks.

Authors’ Response: The experimental results show that several factors, such as strong interaction with Wuhan and high population density, influence the prediction results to some extent. Actually, the spread of COVID-19 in a city is highly influenced by the control measures, which vary from city to city. If a city adopted strict control measures, the spread of COVID-19 may be much slower and less severe than the predicted results.

Reviewer: 2

Comments to the Author

This is a sound analysis of two publicly available data set focusing on intercity migration in China.

The authors may benefit from a recent paper on migration and covid-19 spread published in April issue of Migration Letters journal. That can be useful to better frame the context of this paper indicating wider link between human mobility and disease diffusion.

Authors’ Response: This work was completed on February 19, 2020 (medRxiv 10.1101/2020.02.18.20024570). We used a short historical epidemic spreading data and migration data to develop the model and the corresponding system identification algorithm. At the time of performing this work, there was no attempt in combining SEIR model, migration data and system identification techniques to analyze and predict the spread of COVID-19. The results thus have important indicative values on the effectiveness of using limited initial outbreak data in predicting pandemic progression. Remarks have been added to the Discussions section to highlight this.

Authors may revisit the sentence in second page: "The COVID-19 outbreak, however, began to occur and escalate in a special holiday period in China (about days surrounding the Lunar New Year), during which a huge volume of intercity travel took place, resulting in outbreaks in multiple regions connected by an active transportation network." Because now we know the virus was out and about in as early as early November.

Authors’ Response: We have checked the literature and available data carefully, and found that the “official” data (up to today from the Chinese National Health Committee) indicated the earliest confirmed case in China being December 8, 2019. Indeed, the spread could have started earlier, but our data analysis could only work according to the official data which showed surges in infected numbers in many Chinese cities beginning mid January, which was the period of “spring rush” in China. We have also edited the text so as to emphasize that we referred to the rapid spread in China which was in the period before the Lunar New Year when huge volume of intercity travel took place.

The data needs to be critically presented; Authors indicate the possibility of incompleteness or inaccuracy of official covid data but it seems they assume Baidu data is free of problems. It is important to note the selectivity bias here. This data is collected by an app, which means there are a lot of questions about its representability. This should be clearly noted so readers can interpret it accordingly.

Authors’ Response: Several works adopted Baidu data to investigate the spread of COVID, which have been cited in the paper. Also, we clarified that the Baidu data were expected to be inexact and served to provide indicative travel volumes which were sufficient for the model fitting. This would serve to alert our readers about this issue.

[14] Chinazzi M, Davis JT, Ajelli M, Gioannini C, Litvinova M, Merler S, y Piontti AP, Mu K, Rossi L, Sun K, Viboud C. The effect of travel restrictions on the spread of the 2019 novel coronavirus (COVID-19) outbreak. Science. 2020 Apr 24; 368(6489): 395-400.

[15] Lai S, Ruktanonchai NW, Zhou L et al. Effect of non-pharmaceutical interventions to contain COVID-19 in China [published online May 4, 2020]. Nature. 2020;10.1038/s41586-020-2293-x. doi:10.1038/s41586-020-2293-x

In the conclusion, authors state "The Coronavirus Disease 2019 (COVID-19) epidemic has hit China hard, 331 and as of February 20, 2020, a total of 74,579 infection cases have been 332 confirmed in China, with death toll reaching 2,119." It is a live incidence but it can be useful if they can include the latest statistics regarding the pandemic while making sure the data and analysis refer to an earlier period.

Authors’ Response: We have revised the Introduction to include the latest worldwide figures while emphasizing that this work was completed on February 20, 2020.

---

## [Decision Letter · Decision Letter 1]

12 Oct 2020

Modeling and Prediction of the 2019 Coronavirus Disease Spreading in China Incorporating Human Migration Data

PONE-D-20-05906R1

Dear Dr. Tse,

We’re pleased to inform you that your manuscript has been judged scientifically suitable for publication and will be formally accepted for publication once it meets all outstanding technical requirements.

Kind regards,

Siew Ann Cheong, Ph.D.

Academic Editor

PLOS ONE

Additional Editor Comments (optional):

Reviewers' comments:

Reviewer's Responses to Questions

**Comments to the Author**

1. If the authors have adequately addressed your comments raised in a previous round of review and you feel that this manuscript is now acceptable for publication, you may indicate that here to bypass the “Comments to the Author” section, enter your conflict of interest statement in the “Confidential to Editor” section, and submit your "Accept" recommendation.

Reviewer #2: All comments have been addressed

Reviewer #3: (No Response)

2. Is the manuscript technically sound, and do the data support the conclusions?

Reviewer #2: Yes

Reviewer #3: Yes

3. Has the statistical analysis been performed appropriately and rigorously? 

Reviewer #2: Yes

Reviewer #3: Yes

4. Have the authors made all data underlying the findings in their manuscript fully available?

Reviewer #2: Yes

Reviewer #3: Yes

5. Is the manuscript presented in an intelligible fashion and written in standard English?

Reviewer #2: Yes

Reviewer #3: Yes

6. Review Comments to the Author

Reviewer #2: Authors addressed the key concerns raised by myself and other reviewer. This revised version is of acceptable quality for publication.

Reviewer #3: In this work, the authors attempt to modify the classic SEIR model of disease propagation to include data on human mobility. Specifically, the new model seeks to incorporate fluctuations in the total population into the SEIR, something that was previously taken to be fixed. This work is clearly timely and important and the approach seems reasonable. The authors have addressed the comments of previous reviewers.

My major complaint is that it would be nice to see some verification of the numbers. Clearly, when this paper was originally written that would not be possible as they were predicting the future, but that is no longer the case. Looking at official case numbers and timelines, it seems the authors have done a reasonable job making predictions, but some quantitative measure of correctness at this point would be both possible, and a nice addition. Otherwise, it is not clear the to the reader whether this model is viable for future outbreaks.

More minor comments follow:

1. Page 2, around line 29 states that that cities far from Wuhan have a linear relationship between # of infections and distance, but on a log plot, that is not particularly clear.

2. Page 4, lines 119-123, the authors have, a the reviewers suggestion, attempted to acknowledge the problems with Baidu data by stating that the m_ij need only be accurate relative to one another, but it is not exactly clear why this is the case, since the absolute numbers are used to make predictions of individuals and there is no immediately obvious scaling factor.

3. Page 4, line 129, "as seen in Figure 3" is floating here and should be deleted

4. Page 5, equation 4. The alpha factor does not exist in this set of equations for S and E, though it does show up in later equations. It is not clear to my why this was omitted.

7. PLOS authors have the option to publish the peer review history of their article (what does this mean?). If published, this will include your full peer review and any attached files.

Reviewer #2: No

Reviewer #3: No

---

## [Editor Report · Acceptance letter]

14 Oct 2020

PONE-D-20-05906R1

Modeling and Prediction of the 2019 Coronavirus Disease Spreading in China Incorporating Human Migration Data

Dear Dr. Tse:

I'm pleased to inform you that your manuscript has been deemed suitable for publication in PLOS ONE. Congratulations! Your manuscript is now with our production department.

Kind regards,

on behalf of

Dr. Siew Ann Cheong

Academic Editor

PLOS ONE